# Maternal Immunization with Adjuvanted Recombinant Receptor-Binding Domain Protein Provides Immune Protection against SARS-CoV-2 in Infant Monkeys

**DOI:** 10.3390/vaccines12080929

**Published:** 2024-08-20

**Authors:** Christopher L. Coe, Francesca Nimityongskul, Gabriele R. Lubach, Kimberly Luke, David Rancour, Fritz M. Schomburg

**Affiliations:** 1Harlow Center for Biological Psychology, University of Wisconsin-Madison, Madison, WI 53715, USA; grlubach@wisc.edu; 2Boost Biopharma, Madison, WI 53713, USAfritz.schomburg@lyticsolutions.com (F.M.S.); 3Intuitive Biosciences, Madison, WI 53717, USA; kluke@intuitivebio.com

**Keywords:** SARS-CoV-2, COVID-19, vaccine, receptor-binding domain, recombinant protein, monkey, antibody, IgG, infant, placenta, neutralization

## Abstract

Maternal vaccinations administered prior to conception or during pregnancy enhance the immune protection of newborn infants against many pathogens. A feasibility experiment was conducted to determine if monkeys can be used to model the placental transfer of maternal antibody against SARS-CoV-2. Six adult rhesus monkeys were immunized with adjuvanted recombinant-protein antigens comprised of receptor-binding domain human IgG1-Fc fusion proteins (RBD-Fc) containing protein sequences from the ancestral-Wuhan or Gamma variants. The female monkeys mounted robust and sustained anti-SARS-CoV-2 antibody responses. Blood samples collected from their infants after delivery verified prenatal transfer of high levels of spike-specific IgG, which were positively correlated with maternal IgG titers at term. In addition, an in vitro test of ACE2 neutralization indicated that the infants’ IgG demonstrated antigen specificity, reflecting prior maternal immunization with either Wuhan or Gamma-variant antigens. All sera showed stronger ACE2-RBD binding inhibition when variants in the assay more closely resembled the vaccine RBD sequence than with more distantly related variants (i.e., Delta and Omicron). Monkeys are a valuable animal model for evaluating new vaccines that can promote maternal and infant health. Further, the findings highlight the enduring nature and safety of the immune protection elicited by an adjuvanted recombinant RBD-Fc vaccine.

## 1. Introduction

Initial concerns about the vertical transmission of SARS-CoV-2 from infected women to their infants during pregnancy or when nursing after birth have fortunately proven to not be a major health issue, but more severe symptomatic infections during pregnancy can have adverse clinical consequences [1,2,3,4,5]. COVID-19 in pregnant women has been linked to an increased risk of gestational hypertension/pre-eclampsia, fetal hypoxia and premature birth [6,7,8]. Maternal infections can also affect placental structure and function, and some young children do show signs of long-term neurodevelopmental effects after prenatal infections [9,10,11,12,13,14]. From a public health perspective, the optimal goal would be to prevent more serious cases through maternal immunization, administered either prior to conception or during pregnancy. Prenatal immunization against several pathogens is already recommended by the Centers for Disease Control to protect mothers and infants in the United States, including with the influenza and tetanus, diphtheria and pertussis (Tdap) vaccines.

In addition to reducing the risk of obstetrical complications and maternal morbidity and mortality, many clinical studies have now documented the fact that there are postnatal benefits of prenatal immunizations against SARS-CoV-2 for the young infant [15,16] The transplacental transfer of maternal antibody provides an extended period of passive immunity and defense that can extend for 6–9 months after birth [17,18,19]. The presence of maternal antibody in circulation ensures additional systemic protection beyond the mucosal barrier and innate immune defenses in the nasal passages and upper airways of the young infant. The amount and effectiveness of the transferred SARS-CoV-2 antibody can be even greater in the event of a mild breakthrough infection in a previously vaccinated pregnant woman [20,21,22,23,24,25]. Hybrid immunity of this type has been found to result in higher antibody levels and more enduring protection for their infants, and SARS-specific IgG1 may exceed 140% of maternal levels at delivery [26,27,28,29]. Similarly, one study demonstrated that a third booster immunization against SARS-CoV-2 in pregnant women who had been vaccinated prior to conception was of further benefit for their infants [30]. Nevertheless, there continues to be a widespread hesitancy among women about receiving vaccines while pregnant, despite many studies documenting their overall safety [31,32,33]. In particular, the dissemination of erroneous information about the possibility of mRNA vaccines becoming incorporated into the somatic genome of the host has created unwarranted concern.

Research with animal models can provide valuable information about the safety and efficacy of new vaccine formulations for promoting maternal and infant health. However, selecting the right animal species is critical for acquiring accurate and relevant results. Primates are a good model for SARS-CoV-2 because they are naturally susceptible, and their immune responses and many aspects of their post-infection pathophysiology resemble infected humans [34,35]. For a study of prenatal immunization, it is also important to consider that the timing of maternal antibody transfer to infants differs across species. Specifically, the commonly studied laboratory mouse and rat primarily transfer maternal IgG to their pups in breast milk [36]. Similarly, maternal antibody is transferred postnatally in many domesticated farm animals, including in lactating cows, pigs and horses. In contrast, monkeys are a better model because the Fc receptor for IgG (FcγR) required for binding and transferring maternal antibody is present on their placenta [37]. Previous studies have documented the fact that maternal antibody is transferred at an increasing rate in the final month of pregnancy, and infant IgG1 levels in monkeys typically reach an equivalence with maternal titers at term [38]. However, the placental transfer rates of antibody to different antigens are known to vary in both monkeys and humans [39].

The following experiment was designed to establish the feasibility of using rhesus monkeys to investigate the prenatal transfer of SARS-CoV-2 RBD-specific antibody from the gravid female to her infant. The biological activity of the maternal antibody present in infant circulation was assessed with an in vitro ACE2-RBD binding inhibition assay, which provides a surrogate indication of in vivo neutralization [40]. In addition, we investigated whether immunization of the mother with RBD-Fc containing protein sequences from two different viral variants would have a differential influence on the activity of the prenatally acquired IgG against six variants, from the ancestral Wuhan to the Omicron variant.

## 2. Methods

### 2.1. Subjects

Twelve adult female and infant rhesus monkeys (*Macaca mulatta*, Indian ancestry) were used in this research. They were from a large breeding colony, which was housed in an indoor facility with standardized husbandry conditions. Research staff wore masks and other personal protective equipment (PPE) to minimize cross-species exposure to pathogens, including airborne respiratory viruses. Five adult females were vaccinated prior to conception; one adult female was immunized in early gestation (64 days ga). Following the preconception vaccinations, the cycling females were bred with a timed-mating protocol to determine date of conception and to verify that gestation lengths remained normal [41]. While pregnant, they were socially housed with another female monkey. After delivery, blood samples were collected from all 12 mothers and infants. The research procedures were approved by the Institutional Animal Care and Use Committee (IACUC L006361, L006578).

### 2.2. Blood Collection

Small blood samples (<4 mL) were collected at 2- to 4-week intervals for 3 months after immunization to assess primary and secondary antibody responses in the monkeys vaccinated twice prior to conception (Appendix A). The pregnant female was immunized only once, and her blood was similarly collected for 3 months to track her antibody response. After delivery, small blood samples (1 mL) were obtained from the 6 infants (5 F, 1 M) and their mothers within 1–2 weeks postpartum, to assess placental transfer of RBD-specific IgG (Appendix A).

### 2.3. Immunization

The female monkeys were immunized with a sterile suspension of adjuvanted recombinant receptor-binding domain-IgG1-Fc fusion proteins (RBD-Fc) containing either ancestral Wuhan or Gamma variant protein sequences (aa331–537) (Appendix A). The 1 mL suspension contained either 25 or 50 µg of RBD-Fc. Details about the synthesis protocol have already been published [42]. Four females received intramuscular injections of recombinant RBD-Fc containing protein sequences from the ancestral Wuhan virus. Two were administered RBD-Fc containing sequences from the Gamma variant, an early mutation identified initially in Brazil before spreading to the United States in 2021 [43,44,45]. To enhance the immune response to the recombinant proteins and to activate antigen-presenting cells, AS03 or aluminum hydroxide (Alhydrogel 2%, Al(OH)_3_) was added to the PBS suspension. Five of the six immunizations were carried out with AS03, a widely used adjuvant, including in influenza vaccines, as well as for vaccine studies in monkeys [46]. It was developed originally by GlaxoSmithKline, and is composed of α-tocopherol, squalene and polysorbate 80 in an oil-in-water emulsion [47]. One female monkey’s immunized preconception was administered as a suspension with alum, a semi-crystalline aluminum adjuvant used in Tdap and pneumococcal vaccines; it has also been used in recombinant-protein vaccine studies with monkeys [48].

### 2.4. Antibody Response

Prior to immunization, females were screened with the Colony Surveillance Assay SARS-CoV-2 kit (CSA, Product 12-1225, Intuitive Bioscience, Madison, WI, USA) to ensure they were seronegative and had not been exposed previously to SARS-CoV-2 (Appendix A). It is a validated kit for determining seropositivity in monkey and human sera. This kit was refined with specimens from hundreds of negative samples to determine a threshold cut point for each antigen (S1, S2, and Nucleocapsid). These criteria were determined initially at a 1:200 serum dilution. In addition, test values for sera from other monkeys pre- and post-immunization were available to provide a background context for the 3 dilutions used in the current research (1:1000, 1;10,000, 1:100,000). The CSA provides a high-sensitivity test for identification of antibodies to the SARS-CoV-2 spike S1 and S2 subunits and nucleocapsid antigens in a multiplex format for simultaneous detection of antigen-specific antibodies in each well [49]. The image-capture system uses a CCD camera to measure the signal on a white-to-black scale, which is quantified as Relative Intensity Units (RIUs). The signal increases with increasing amounts of bound antibody. SARS-CoV-2 recombinant proteins (S1, aa16-685; S2, aa686-1213, and nucleocapsid, aa1-419) expressed from CHO or HEK cells are used in the assay to ensure proper glycosylation. Antibodies to spike S2 and nucleocapsid were not detected after immunization in any monkey; therefore, this report focuses on IgG responses to only the spike S1 subunit. Sera from the immunized females, and then from mothers and infants after birth, were tested at 3 dilutions: 1:1000, 1:10,000, and 1:100,000. Validated Negative sera (pre-2019 Specific-Pathogen-Free blood from rhesus macaques) and Positive sera (from SARS-CoV-2 infected monkeys) were run on every plate. The 1:200 dilution yields a mean 15,525 Relative Intensity Units (RIUs) for S1, with a minimum mathematical value of 0 and a maximal RIU value of 65,555. However, threshold cutoffs are influenced by the dilution. At the 1:10,000 serum dilution, the threshold cutoff for seropositive is >1000 RIU. Sera with RIU values under 1000 are classified as seronegative.

### 2.5. ACE2 Neutralization Assay 

Maternal and infant sera were also evaluated with a COVID-19 Neutralization assay (Meso Scale Discovery, MSD, Rockville, MD, USA), which quantifies how much the antibody in the specimen blocks the binding of recombinant ACE2 with SARS-CoV-2 SP and RBD antigens. Sera were run at 1:100 using the kit diluent and tested against an antigen panel with 6 viral variants (Wuhan, Alpha, Beta, Gamma, Delta, and Omicron) (MSD Panel #K15562U-2). Percent inhibition (%) was calculated by comparing the inhibition elicited by sera with the effect of diluent alone. Details about the specificity and sensitivity of this platform have been reported previously [40,50]. In brief, this assay utilizes SARS-CoV-2 RBDs immobilized in microarray format, which are then incubated with test sera in solution, followed by incubation with labelled ACE2 receptor. If an interfering molecule is present in the sera, such as RBD-specific IgG1, it inhibits binding of RBD to ACE2, and a decrease in signal is observed. Following the manufacturer’s instructions, the assay diluent is used as the primary Negative assay control, run in duplicate determinations, and its low binding inhibition is used as the reference point when calculating the percent inhibition exhibited by the monkeys’ sera. BI (%) is calculated with the following formula: 1 − (Experimental Sample/Assay Diluent) × 100. To provide additional Negative Controls, we included the pre-immunization baseline samples from the adult females, as well as sera from 6 additional seronegative adult monkeys who had not been immunized or exposed to SARS-CoV-2. To serve as Positive Controls, our assay protocol included anti-SARS-CoV-2 spike RBD neutralizing antibody (Human IgG1, ACRO Biosystems, SAD-S35, Lot # S35-211VF1-VT run at 1:100 and 1:1000, 1.0 µg/mL and 0.1 µg/mL, respectively), and a pooled human SARS-CoV-2 national IgG positive standard (Frederick National Laboratory Lot # COVID-NS0109, run at 1:10, 1:100 and 1:1000). The Positive Controls were included to validate the assay’s performance and were not used to calculate the binding inhibition of the experimental samples.

### 2.6. Comparison with Placental Transfer of Maternal Antibody to Tetanus Toxoid (TT)

After the final 3-month blood sample to track the monkeys’ response to RBD-Fc immunization, each female was administered a pediatric vaccine with tetanus toxoid (Dtap, Deptacel, 0.1 mL IM, 1/5 human dose). For the 5 females immunized prior to conception, the Dtap was injected prior to breeding and conception, whereas for the pregnant female, it was administered in late gestation. The purpose was to compare the placental-antibody transfer rate for a different antigen. Anti-tetanus toxoid IgG1 levels were quantified in the postpartum samples with the Monkey Tetanus Toxoid ELISA kit (Creative Diagnostics, Shirley, NY, USA). Maternal and infant sera were run in duplicate determinations at two dilutions (1:200 and 1:400) and referenced to a serially diluted anti-TT IgG1 calibrator. Placental transfer rate of TT-specific antibody was calculated as the infant IgG1/maternal IgG1 ratio.

### 2.7. Statistical Analyses

Before conducting statistical tests, descriptive statistics, including mean and variance, were examined, as well as verifying the absence of extreme outlier values. The antibody responses of the adult females to RBD-Fc immunization were evaluated with a repeated-measures analysis of variance (ANOVA) to assess temporal changes. IgG levels in the mothers and infants after birth were compared with a two-factor ANOVA to analyze placental transfer and determine the effect of prior immunization history with either Wuhan or Gamma RBD-Fc. Correlations between maternal and infant IgG levels were evaluated with Pearson’s *r* tests. Differences in ACE2-RBD binding inhibition were interrogated with two-factor ANOVAs, including host age (mother or infant) and antigen used for immunization (Wuhan or Gamma RBD-Fc) treated as between-subject factors.

## 3. Results

### 3.1. Pregnancy Outcomes

All six infants (5 F, 1 M) were born full-term from unassisted deliveries. Gestational length was not affected by the immunizations and was in the normal range (mean = 168.5 days, 166–172 days). Similarly, all birthweights were typical for a rhesus monkey neonate at term (mean = 472 g, 442–521 g).

### 3.2. Antibody Response to Immunization

The adult monkeys were screened for background antibody prior to immunization with the recombinant RBD-Fc. All six females were verified to be seronegative for SARS-CoV-2 S1, S2, and nucleocapsid, prior to administration of the vaccine. RIU values were below 1000, the cutoff level for seropositivity. After injection of the adjuvanted recombinant RBD-Fc, antibody to the S1 subunit RBD was detected at 2 weeks. The levels then increased above the criterion for seropositivity by 4 weeks (>1000 RIU for the 1:10,000 dilution). The booster immunization with RBD-Fc at 1 month elicited further increases in antibody to the S1 RBD subunit, but the monkeys continued to remain seronegative for both S2 subunit and nucleocapsid antigens. Figure 1 shows that antibody levels to the Spike 1 subunit RBD increased significantly over time, with maximal levels peaking at 2 weeks after the booster (*F*{5,20} = 73.5, *p* < 0.001). Antibody values are expressed in relative intensity units (RIUs) and illustrated for the 1:10,000 serum dilution. During the secondary response, antibody could be detected even in extremely dilute sera, at 1:100,000.

### 3.3. Placental Transfer of Maternal IgG to RBD-Fc

The female monkeys continued to have high levels of IgG specific to the RBD of the S1 subunit after delivery and remained seronegative for S2 subunit and nucleocapsid antigens (Appendix A). High levels of IgG specific to the S1 subunit RBD were similarly present in infant circulation, and did not differ significantly from maternal values at term. Figure 2A shows the equivalence of maternal and infant antibody level at each of the three serum dilutions tested in the assay (1:1000, 1:10,000, and 1:100,000). A scatterplot showing the positive correlations between maternal and infant values at each dilution is illustrated in Figure 2B. The overall correlation between mother and infant antibody levels was significant across the three dilutions (*r*{18} = 0.98, *p* < 0.001). The similarity in the RBD-Fc-specific IgG levels was also evident when considering the placental transfer rate, which averaged 100.3%. The adult female monkey immunized during early pregnancy also had high levels of spike-specific antibody at delivery, which were in the same range as the postpartum antibody of the five other females immunized prior to conception. The amount of RBD-Fc-specific IgG present in her infant indicated there was also a typical placental transfer rate of 101.7%. While there were high circulating levels of IgG for the RBD of the S1 subunit in the six infants, all were seronegative for S2 subunit and nucleocapsid antigens (Appendix A). The concordance of maternal and infant antibody levels seen for the RBD of the S1 subunit was also not evident in the assay values for the S2 subunit or nucleocapsid antigens.

### 3.4. ACE2 Neutralization

Both maternal and infant sera were highly effective at inhibiting the binding of soluble ACE2 protein to the RBD coated on the plate wells in the in vitro neutralization assay (Figure 3). However, the extent of the inhibition was strongly influenced by the type of RBD-Fc administered to the female monkey when immunized. Antibody transferred prenatally to the infants evinced the same pattern of immunogen specificity, with more binding inhibition of the RBD-Fc to immobilized ACE2 evident for the variant to which their mother had been immunized.

A significant interaction term in the ANOVA comparing the type of RBD-Fc used to immunize the females and the extent of inhibition of ACE2-RBD binding with Wuhan or Gamma affirmed the concordance between immunogen and the target variant in the assay (*F*{1,10} = 79.03, *p* < 0.001). In addition to considering functional activity with respect to Wuhan and Gamma variants, the ACE2 neutralization test was used to examine ACE2-RBD binding inhibition with four other SARS-CoV-2 variants (Appendix A). Sera from both mothers and their infants showed more biological activity against early variants, with significantly less ACE2-RBD neutralization evident for the Delta and Omicron variants (*F*{5,40} = 23.31, *p* < 0.001).

### 3.5. Placental Transfer of IgG1 against Tetanus Toxoid (TT)

Postpartum sera from both mothers and infants also had high levels of IgG1 against tetanus toxoid (TT) (1.29 (0.38) and 1.01 (0.36) µg/mL, respectively). However, the placental transfer rate of TT antibody was only 78% (S.D. = 29%), lower than the transfer rate of 100.3% (S.D. = 20%) for the maternal RBD-Fc-specific IgG present in infant circulation. Despite the difference in placental transfer rates, there was a positive association between the RBD-specific IgG and TT-specific IgG levels in the six infants (*r* = 0.80, *p* < 0.06), as well as a significant correlation between the placental transfer rates of RBD-specific and TT-specific IgG (*r* = 0.87, *p* < 0.02).

## 4. Discussion

In keeping with previous research, our results showed that adjuvanted recombinant-fusion-protein formulations are highly immunogenic in nonhuman primates, and these vaccines have been found to elicit vigorous antibody and T-cell-mediated responses to SARS-CoV-2 antigens in both monkeys and humans [51,52,53]. While some previously used RBD-based constructs were similar to the vaccine administered in the current study, the RBD-Fc fusion protein used to immunize the adult females contained additional residues of the C-terminal S1 domain, which may facilitate immunogenicity, either directly by providing more potential epitopes, or indirectly by enhancing the stability of the RBD-Fc construct and conformational flexibility. Our C-terminally extended RBD construct dimerizes through the disulfide bridge present in the Fc portion, which may also reveal non-linear epitopes in its dimerized form.

Another novel aspect of this study was demonstrating that spike-specific antibody in the gravid female is transferred to the fetal monkey before term, and high circulating levels of RBD-Fc-specific IgG are present in the neonatal monkey after birth. This finding in monkeys concurs with many large clinical studies that have documented the transfer of maternal antibody from pregnant women to their infants, both after SARS-CoV-2 infections during pregnancy and following vaccination against COVID [54,55]. In addition, serum from infant monkeys was found to evince the same capacity as maternal sera to inhibit viral RBD binding to ACE2 in an in vitro neutralization assay. The infants’ sera also revealed immunogen specificity, reflecting whether the adult female monkeys had been immunized with RBD-Fc containing protein sequences from the Wuhan or Gamma variants. Specifically, the placental transfer of maternal antibody after immunization with Wuhan RBD-Fc enabled infants to more actively inhibit ACE2-RBD binding with the ancestral variant, whereas sera from the two infants born to mothers immunized with Gamma RBD-Fc exhibited more inhibition of ACE2 binding with Gamma. Further, the sera from all mothers and infants elicited less inhibition of ACE2-RBD binding when tested against later sequence-divergent variants: Delta and Omicron. As other vaccine studies with monkeys and humans have found, it will continue to be challenging to produce variant-specific vaccines that provide optimal cross-protection against new variants, with many mutations that enable immune evasion [46].

The findings on the high placental transfer rate of RBD-Fc-specific antibody also concur with previous research showing that infant IgG levels in the rhesus monkeys typically attain equivalence with maternal levels by term. Comparisons of different species across the order Primates have shown that the transfer of maternal antibody switches from a postpartum process in breast milk in the ancestral prosimians to a prenatal pathway in Old World monkeys and apes, coinciding with the evolutionary emergence of the FcγR on the monkey and ape placenta, which binds and transfers maternal IgG to the fetus [36]. Like human pregnancies, maternal IgG, primarily of the IgG1 subclass, is transferred in an accelerating manner in the final weeks of pregnancy [56]. Therefore, an infant born premature is deprived of this important endowment of maternal immunity and may be more vulnerable to a postnatal infection [55]. Virus-induced damage to the placenta can also reduce the mother–infant antibody transfer rate, even though spike-specific IgG continues to remain high in the human neonate after maternal SARS-CoV-2 infections [57].

It was also of interest that the placental transfer rate found for SARS-CoV-2 RBD-specific antibody was higher than for anti-TT IgG1, given that anti-TT antibody readily transfers to the fetal compartment after pregnant women are administered Dpat vaccines [36]. With some antigenic and pathogen exposures, monkeys do exhibit a more active transfer rate, such as when pregnant rhesus monkeys were infected with influenza virus in mid-gestation. Flu-specific IgG levels in the infant after birth were 20% higher than maternal IgG levels at term [58]. More systematic research is still needed to better understand the factors that account for the differential transfer of antibody to various antigens. Further, having demonstrated the value of nonhuman primates for this type of investigation, it would be important to also examine the influence of the neonatal Fc receptor for IgG (FcRn) in infant monkeys, because it can affect how long the maternal antibody remains in circulation [59]. In humans, the half-life of maternal IgG is approximately 3 weeks, which accounts for the decline in maternal antibody and immune protection by 6–9 months postpartum [60].

Notwithstanding the consistency of the current results, it should be acknowledged that the sample size was small. The current results will have to be replicated and extended. Our experiment was conducted primarily to establish feasibility and proof of principle that monkeys can serve as an appropriate animal model for investigating the placental transfer of maternal antibody against SARS-CoV-2. The novel observation that ACE2-RBD binding inhibition was differentially influenced by immunization with RBD-Fc fusion proteins containing specific sequences from the Wuhan and Gamma variants will also have to be verified. While preliminary, this finding would have important implications as SARS-CoV-2 continues to mutate. Vaccine formulations administered to women of child-bearing age will need to be targeted to the SARS-CoV-2 variants currently in circulation, to be of optimal benefit for their infants [61]. We were also not able to consider the relative immune enhancement elicited by different adjuvants in the current experiment. The antibody levels elicited in the one monkey administered recombinant RBD-Fc in an alum suspension was similar to the responses of the other five females. However, we have compared the response to AS03 and alum adjuvant vaccinations in 12 nonpregnant monkeys. Based on those unpublished results, primary antibody levels are initially similar, but there is a tendency for a more robust secondary response after a booster immunization with AS03. The current experiment was also not powered or intended to consider the influence of fetal sex on the placental transfer of maternal antibody. Based on prior research in rhesus monkeys and other primates, it is unlikely that there would be a large difference in the amount of maternal IgG transferred to female and male infants [62,63]. However, this question should be investigated further, because following maternal infections with SARS-CoV-2, one study found significant effects of fetal sex on the extent of inflammatory activity and placental damage, which then impacted the amount of maternal antibody transferred to the developing infant [64]. Finally, it should be acknowledged that the infants’ blood was collected only once after birth. Future research will need to serially track how rapidly the levels of maternal IgG1 in infant circulation decline over time. It was striking that neonatal sera exhibited such a high level of ACE2-RBD binding inhibition, but a virus challenge protocol is required to ascertain if the declining levels of neutralizing antibody in an older infant would provide effective protection against a productive viral infection.

## 5. Conclusions

Administration of adjuvanted recombinant-protein antigens comprised of SARS-CoV-2 RBD-human IgG1-Fc fusion proteins (RBD-Fc) containing either ancestral Wuhan or Gamma protein sequences was found to be highly immunogenic in female monkeys and facilitated the placental transfer of RBD-Fc-specific antibody to their infants. The immune response after immunization preconception proved to be sufficiently enduring to ensure maternal antibody levels were maintained at a high level through the 5.5-month pregnancy to term. The sustained response may reflect the fact that memory B cells in monkeys continue to produce SP-specific antibody for longer than the typical humoral response in humans [65,66,67]. The IgG synthesized by monkey lymphocytes can even show a gain in neutralizing efficiency over time [68]. Importantly, based on the ACE2-RBD binding inhibition assay, the antibody in infant circulation showed neutralizing activity comparable to adults. It also evinced immunogen specificity, reflecting whether the mother had been immunized with protein sequences from the ancestral Wuhan or Gamma variant. The variant-specific activity affirms the cautionary concern that immunization protocols for women of child-bearing age will require antigens from variants currently in circulation, to be of optimal benefit for their infants [69]. Having verified that the rhesus monkey can provide an appropriate animal model for investigating the transfer of protective maternal antibody, future studies can address other critical issues, including vaccine safety [70], the optimal timing of maternal immunization for protecting preterm and term infants [71] and the refinement of adjuvant formulations for use with recombinant-protein vaccines. In addition, while studies have indicated that the placentally transferred antibody is predominantly IgG1, it will also be important to conduct a subclass analysis, especially after immunizations during pregnancy, because influenza vaccinations in late gestation were found to differentially affect the transfer rate of the IgG3 subclass to infants [72].

**Summary Sentence**. Immunization of female rhesus monkeys with adjuvanted recombinant SARS-CoV-2 RBD-IgG fusion proteins resulted in the transplacental transfer of maternal RBD-specific IgG to their infants, which enabled the infants’ sera to actively inhibit ACE2-RBD binding.

## Figures and Tables

**Figure 1 vaccines-12-00929-f001:**
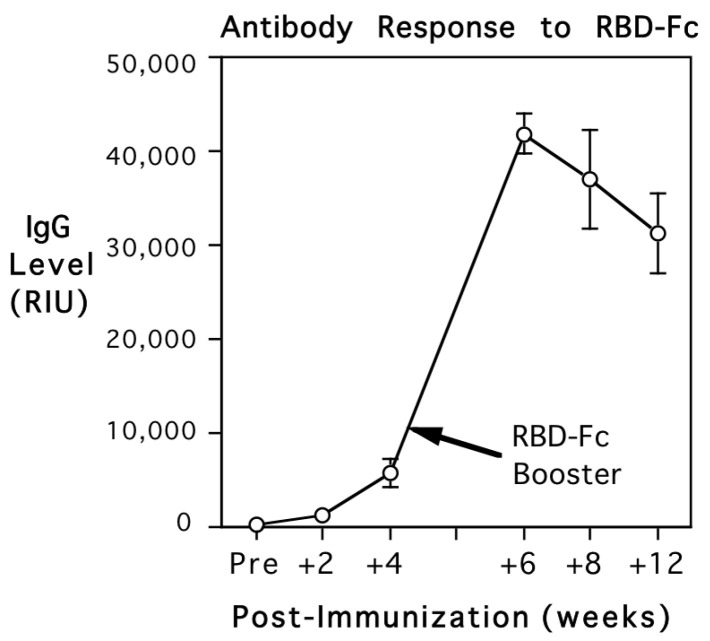
Primary and secondary antibody responses of 5 adult female rhesus monkeys after immunization with adjuvanted recombinant spike protein-fusion protein (RBD-Fc) prior to conception. Mean (S.E.) antibody titers are shown in relative intensity units (RIUs) for the 1:10,000 serum dilution. Monkeys were screened at baseline to verify they had not been exposed previously to SARS-CoV-2. IgG levels increased significantly over time, crossing the threshold for seropositivity.

**Figure 2 vaccines-12-00929-f002:**
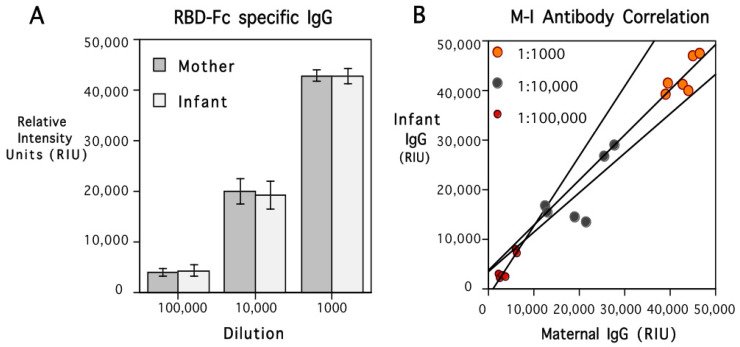
Mean (S.E.) spike protein-specific IgG levels in 6 mother and infant rhesus monkey pairs after delivery are shown in (**A**) (dark and light bars, respectively). Antibody titers were quantified at 3 serum dilutions (1:1000, 1:10,000, 1:100,000), and values are expressed in relative intensity units (RIUs). Maternal and neonatal antibody levels were similar at each dilution, resulting in a mean placental transfer rate of 100.3%. The ANOVA indicated only a significant effect of serum dilution (*F*{2,15} = 133.1, *p* < 0.001). Individual data for all infants and the correlation with their mothers’ IgG titer at each dilution are shown in (**B**). Maternal and infant IgG levels were similar at each dilution, positively correlated at each dilution (*r* = 0.77, *p* < 0.08; *r* = 0.74, *p* < 0.09, and *r* = 0.94, *p* < 0.01, respectively), and significantly correlated across the entire series (*r*{18} = 0.98, *p* < 0.003). Regression lines are plotted to show the mother–infant association at each dilution. Placental transfer rate was calculated to be 100.3% and was markedly higher than the transfer rate of IgG transfer for tetanus IgG1, which was only 78%.

**Figure 3 vaccines-12-00929-f003:**
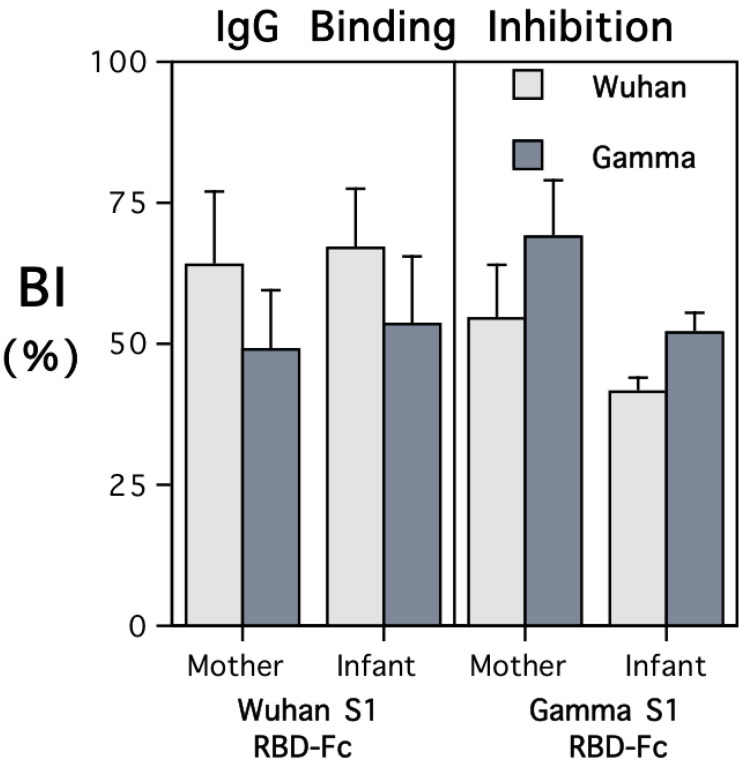
ACE2 binding inhibition by sera from mothers and young infants after delivery. The results are shown separately for the females who had been immunized with recombinant RBD-Fc including protein sequences from the S1 subunits of the Wuhan or Gamma variants. The viral variant tested in the neutralization assay is identified by light and dark shading of the bars (Wuhan and Gamma, respectively). The pattern of binding inhibition revealed immunogen specificity. The infants’ sera also evinced more bioactivity against the variant against which the adult females had been immunized. As a Negative assay control condition, non-immunized sera from 12 seronegative female monkeys were also tested and elicited minimal binding inhibition (from 0 to <10%). Further, the sera from the immunized mothers and their infants elicited more binding inhibition than the two Positive assay controls at the dilutions tested. For monkeys immunized with RBD-Fc containing Wuhan protein sequences, the range of BI values was 35.8–99.3; for those immunized with RBD-Fc containing Gamma protein sequences, the range was 26.3–86.8. A graph showing the ACE2 neutralization results for 6 viral variants, including Delta and Omicron, is provided in Appendix A.

## Data Availability

Requests by professional scientists or clinical investigators for access to original data should be directed to the first author (C.L.C.).

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
