# Peer review of "Maternal Immunization with Adjuvanted Recombinant Receptor-Binding Domain Protein Provides Immune Protection against SARS-CoV-2 in Infant Monkeys"

_vaccines, 2024, doi:10.3390/vaccines12080929_

Round 1

Reviewer 1 Report

Comments and Suggestions for Authors

Review report on the article entitled “Maternal immunization with adjuvanted recombinant receptor-binding domain  protein provides immune protection against SARS-CoV-2 in infant monkeys”

The authors evaluated the potential of using an animal model (monkeys) to test and validate if a vaccine to SARS-CoV-2 can induce placental transfer of maternal antibody. This study showed that there is a correlation between the serum antibody levels of mother and infant at birth and the vaccination before and during pregnancy did not induce adverse effects on the pregnancy itself. This paper is well written and well-presented and will be of general interest to researchers in the field.

Listed below are some comments and suggestions for its improvement.

Section 2.1, Line 89:  Add exact number of animals used.

Section 2.3, Lines 115-118: Be more specific - which formulation contained the aluminum and which the ASO3 adjuvants. Also, in the results or discussion section there were no description as to which of these two adjuvants produced the highest antibody tires or if any variations were observed between the two? Please comment on this in at least the discussion section.

Section 2.4: Line 135: Please include what are the maximum RIU level considered to be negative, and the minimum level considered positive.

Section 2.5 Lines 150-53: The positive control results needs to be included as part of the results and indicate what was used as a negative control?

Section 2.6 line 157: Please indicate if this vaccine was administered before or during pregnancy.

Section 3.1, Line 178: Exact number (n=6) to be included

Figure 1: Please indicate in the legend of average of the 6 were used. Or is this the data for one animal? Also indicate: Were the same profile seen for those vaccinated with the different strains and with the two adjuvants used? I.e is the figure representative of one vaccine formulation or combined for all irrespective of the vaccine formulation?

Figure 2: Include positive and negative results.

Section 3.3 Line 204: Please include Supplementary Materials (Figure S1).as part of the main paper – this is important results that should not be shown as supplementary.

Author Response

Section 2.1, Line 89: Add exact number of animals used.

Number of animals assessed in this study has been added. The N is 12 (6 mothers and 6 infants).

Section 2.3, Lines 115-118: Be more specific - which formulation contained the aluminum and which the ASO3 adjuvants. Also, in the results or discussion section there were no description as to which of these two adjuvants produced the highest antibody titers or if any variations were observed between the two? Please comment on this in at least the discussion section.

Details about the AS03 and alum adjuvants have been added to both Methods and Results. In addition, we now acknowledge that the differential effectiveness of the two adjuvants will have to be addressed in future research. We do have unpublished data on 12 monkeys (6 administered RBD-Fc with AS03 and just one injected with an alum suspension) and have provided this information in the Acknowledgements paragraph of the Discussion. Within the context of the current experiment, there wasn’t a difference in either the antibody levels or amount of ACE-2 binding inhibition.

Section 2.4: Line 135: Please include what are the maximum RIU level considered to be negative, and the minimum level considered positive.

We now provide the cutoff threshold value for seronegative and seropositive. In brief, at the 1:10000 serum dilution, antibody levels below 1000 RIU are considered to be negative, whereas the criterion for seropositive is a RIU >1000. After the booster immunization, the RIU values at the 1:10,000 dilution were above 40,000 during the secondary response.

Section 2.5 Lines 150-53: The positive control results needs to be included as part of the results and indicate what was used as a negative control?

We have now provided more details about the experimental protocol and assays, including the inclusion of both negative and positive controls in both assays. The values for the controls are provided in the Figure Legends as well as in the new supplemental table (Figure S1).

Section 2.6 Line 157: Please indicate if this vaccine was administered before or during pregnancy.

We have clarified that this figure shows the antibody responses of the 5 females immunized prior to conception. Within the text of the Results, we report that the antibody levels of the one female immunized in early gestation were similar at term.

Section 3.1, Line 178: Exact number (n=6) to be included

We have clarified that the N for the maternal sera is 6, and similarly the N for the infant sera is also 6.

Figure 1: Please indicate in the legend of average of the 6 were used. Or is this the data for one animal? Also indicate: Were the same profile seen for those vaccinated with the different strains and with the two adjuvants used? I.e., is the figure representative of one vaccine formulation or combined for all irrespective of the vaccine formulation?

We have clarified in the Figure Legend that this graph shows the antibody responses of the 5 females immunized prior to conception. Within the text of the Results, we report that the antibody levels of the one female immunized in early gestation were similar to the other females at term.

Figure 2: Include positive and negative results.

Because we were also asked to include the supplemental figure (the previous Figures S1) within the article, and Figure 2 is now a 2-panel graphic, it would be too cluttered and confusing to include the positive and negative assay control values within the graphic. However, we discuss the values for the negative and positive controls in the new Figure Legend for Figure 2A and B. As noted above, the negative sera have RIU values below 1000, often a value of 0. Conversely, a sample was considered to be seropositive if the RIU value is above 1000 but the values often reach 40000-60000 depending on the serum dilution. We also modified how we show the error bars for the bars with the mean values in Figure 2. We now show both the positive and negative variance around the mean values.

Section 3.3 Line 204: Please include Supplementary Materials (Figure S1) as part of the main paper – this is important results that should not be shown as supplementary.

As requested, Figure S1 is now presented within the main part of the report. It is included as the new Figure 2A. At the same time, we created a new table for the Supplementary Materials, which provides more information on the negative and positive controls used in the CSA kit to quantify the antibody levels. In addition, Table S1 affirms the specificity of the antibody response to the S1 subunit but showing absence of an antibody response to the S2 subunit and nucleocapsid antigens in the immunized mothers and their infants.

Reviewer 2 Report

Comments and Suggestions for Authors

The authors have submitted the manuscript titled "Maternal immunization with adjuvanted recombinant receptor-binding domain protein provides immune protection against SARS-CoV-2 in infant monkeys". In this, the authors investigated the feasibility of using non-human primates to study the placental transfer of maternal antibodies against SARS-CoV-2. 

I have described my concerns about the manuscript below:

1. The major thing missing here is a negative control. While authors have shown the data from the female monkeys prior to immunization, they do not have a non-vaccinated female monkey in the study. Thus, there is no negative control for the infant monkeys' data. Without the appropriate negative control, any kind of statistical comparison become irrelevant to draw any conclusions.

2. The authors should specify the sex of the infant monkeys. Although it is not possible to determine the placental transfer of antibodies as a function of sex of the infants here due to the small number of animals, it should be acknowledged that this could have influenced the results (If they don't have 50% males and females in the infant monkeys).

Comments on the Quality of English Language

There are a few typos in the manuscript that require thorough proofreading. Other than that, the English language has no issues.

Author Response

  1. The major thing missing here is a negative control. While authors have shown the data from the female monkeys prior to immunization, they do not have a non-vaccinated female monkey in the study. Thus, there is no negative control for the infant monkeys' data. Without the appropriate negative control, any kind of statistical comparison become irrelevant to draw any conclusions.

    We are grateful for the time and effort invested by the reviewers to evaluate our submission to Vaccines. We appreciate that the merits of the scientific question were recognized while concerns were expressed about additional points that needed clarification and several issues that need to be investigated and addressed in future studies. We agree that it is important to acquire more information on the relative effectiveness of different adjuvants when administering adjuvanted recombinant proteins. In addition, it is clinically important to determine if there is a difference in the extent of immune protection provided to female and male infants via prenatal immunizations. The current experiment was conducted to verify the feasibility of refining this type of nonhuman primate model and establishing proof of concept to provide a foundation for these types of additional studies by us and others. Given the cost of primate research and the time needed to conduct longitudinal assessments from the preconception immunization to birth, which extends beyond a year given the long gestation in monkeys, we first needed to document the potential value and utility of a primate model.

    In the revision, we provide the additional information requested by the 2 reviewers about the experimental protocol. We also expanded the Caveats and Limitations paragraph in the Discussion section, acknowledging that there are several other important issues that need to be addressed in future research. In some instances, we already had information from other experiments, such as whether the amount of maternal IgG transferred prenatally to fetal monkeys differs between females and males in the rhesus monkey. Both in the manuscript and below, we have provided more information about how the analysis was controlled and standardized to ensure that the antibody levels measured in the mothers and infants reflected the response to the RBD-Fc used for immunization and that it was not elicited by inadvertent exposure to virus. In addition, we have provided the requested information about the controls in the assays, which did include both seronegative and seropositive sera. As requested, we have also provided the threshold cutoff values for categorizing an experimental sample as seronegative or seropositive.

    The additional information and changes are detailed below.

    Experimental Controls and Standards

    We were remiss in not more clearly communicating that the experiment and analysis did include important controls. The 6 adult monkeys were verified to be seronegative at baseline prior to the immunization and had not been exposed previously to SARS-CoV-2 antigens. No monkey in this colony has ever been infected with SARS-CoV-2. Although only these 6 female monkeys were bred and have birthed infants so far, we have collected serial samples from other adult monkeys. Unless immunized with recombinant RBD-Fc, none ever spontaneously seroconverted. It is also important to emphasize that we were also looking for specificity in the antibody responses to the S1 subunit. When each serum sample was assayed, we simultaneously determined if there was antibody present for the S2 subunit or nucleocapsid antigens. Maternal and infant samples evinced antibody only to the S1 subunit, and the levels of antibody in all of the blood samples exceeded the threshold for seropositivity. In addition, these samples had antibody values below the positive cutoff level for the SARS-CoV-2 S2 subunit and nucleocapsid antigens. To share this information with the reviewers and journal readers in a forthcoming manner, we created and added a new table in Supplementary Materials. This table shows the values exactly as they were generated by the CSA kit. In addition, we did not convey clearly enough that both Negative and Positive Controls were included in each assay run. The new Table S1 also provides the values for the control sera in the assay.

    Similarly, there were both negative and positive controls when running the MesoScale Discovery ACE2 neutralization assay. In the same assay that generated the binding inhibition (BI) data for the mother and infant samples, we also included sera from the pre-immunization baseline samples of the adult females. Those samples elicited only minimal inhibition, which was really at the background level for this assay (<10%). As negative controls, we also include sera from 6 additional monkeys that were seronegative and never immunized. They also did not result in significant BI. Further, the manufacturer instructions indicate that the assay diluent run neat should be considered as the negative control, which was also done. There were two positive controls included in the BI assay as well.

  2. The authors should specify the sex of the infant monkeys. Although it is not possible to determine the placental transfer of antibodies as a function of sex of the infants here due to the small number of animals, it should be acknowledged that this could have influenced the results (if they don't have 50% males and females in the infant monkeys).

    Sex of Infant

    As requested, we added the sex distribution of the infants to the Methods section (Section 2.5, Line 106). There were 5 females and 1 male. The small N and uneven sex distribution precluded statistical comparisons, but we know from prior research on a larger number of infant monkeys that there isn’t a significant influence of fetal sex on placental transfer rates in the rhesus monkey (Coe et al., 1993). We have also considered the question of an influence of fetal sex on placental transfer of maternal IgG in another monkey species, the squirrel monkey. There also wasn’t a difference in IgG levels in female and male infants at birth in squirrel monkeys, unless one purposefully sought to perturb placental antibody transfer via sustained stressful manipulations of the gravid female (Coe & Crispen, 2000). But we do now acknowledge the variable of fetal sex as a caveat/limitation in the Discussion section because it could prove to be important with respect to a differential vulnerability of the female and male fetuses to maternal infection and potentially could influence the duration of the postnatal immune protection in developing infant. Supporting the importance of this hypothesis, we added a citation to a publication on human infants, which identified a differential effect of maternal SARS-CoV-2 infection on the placentae of male and female fetuses, which then affected the placental transfer of maternal antibody (Bordt et al., 2021).

    Coe, C.L.; Kemnitz, J.W.; Schneider, M.L. Vulnerability of placental antibody transfer and fetal complement synthesis to disturbance of the pregnant monkey. J. Med. Primatol. 1993, 22(5), 294-300.

    Coe, C.L.; Crispen, H.R. Social stress in pregnant squirrel monkeys (Saimiri boliviensis peruviensis) differentially affects placental transfer of maternal antibody to male and female infants. Health Psychol. 2000, 19(6), 554-559.

    Bordt, E.A.; Shook, L.L.; Atyeo, C.; Pullen, K.M.; De Guzman, R.M.; Meinsohn, M.-C.; Chauvin, M.; Fischinger, S.; Yockey, L.J.; James, I.; Lima, J.R.; Yonker, L.M.; Fasano, A.; Brigida, S.; Bebell, L.M.; Roberts, D.J.; Pepin, D.; Huh, J.R.; Bilbo, S.D.; Li, J.Z.; Kaima, A.; Schust, D.J.; Gray, K.J.; Lauffenburger, D.; Alter, G.; Edlow, A.G. Maternal SARS-CoV-2 infection elicits sexually dimorphic placental immune responses. Sci. Transl. Med. 2021, 13, eabi7428. doi:10.1126/scitranslmed.abi7428

    We hope that the additional information and the changes in the figures and presentation have improved our submission. We tried to be responsive to the requests by both reviewers and the constructive criticism, which guided us on how to improve.

Reviewer 3 Report

Comments and Suggestions for Authors

The present study aims to determine the efficacy of a maternal vaccination strategy using a non-human primate model. To this end, the authors used adjuvanted recombinant protein antigens consisting of human IgG1-Fc combined with the receptor binding domain (RBD) of the spike protein of SARS-CoV-2 as a vaccine in immunized pregnant rhesus macaques. Antigen-specific IgG antibody responses and their neutralizing activities of ACE2 binding were evaluated. The results are intriguing; however, the authors need to address several points listed below to improve the quality of the manuscript.

Major points:

1.    It is important to show how long dose maternally transferred antigen-specific IgG antibodies persist in the infant monkey with neutralizing function. 

2.    The authors should determine the subclass of antigen-specific IgG antibodies in mothers and infants.

3.    For the neutralization assay, it is essential to have a positive control known to have significant neutralizing activity.

Minor point:

Although the authors have already shown their work, it would be nice to have illustrations of the vaccine antigen construct and the immunization and sampling schedule.  

Author Response

Review 3

We appreciate that third reviewer found this research topic to be important and that the findings from our experiment to establish feasibility were ‘intriguing’.  We agree with his/her view, as well as the opinions of the other 2 reviewers, that there are many important issues and specific details that now need to be resolved further.

We have acknowledged several of these concerns as limitations in the Discussion section. Even though it is not possible to acquire more samples from these particular mothers and infants at this point, some of the reviewer’s concerns and questions can be addressed by reference to the published literature. 

Major points:

  1. It is important to show how long dose maternally transferred antigen-specific IgG antibodies persist in the infant monkey with neutralizing function. 

Unfortunately, we cannot collect new samples from these infants at this point because they now are much older. However, in the Discussion section, we do provide information about the duration of passive immunity. With a half-life of 3 weeks for maternal IgG in infant circulation, and thus decreasing levels of maternal IgG over time, one would expect to see a parallel decline in ACE2 neutralization.

A primary purpose of our feasibility experiment was to establish proof of principle so that it would provide the foundation for the type of longitudinal study the third reviewer would like to see.  Given the long duration of pregnancy in monkeys (5.5 months) and the length of the passive immune phase (6-9 months), it would require a new study that lasts 1.5 years. Because of the cost of research with nonhuman primates, a project of that type will require the receipt of a substantial award.  Hopefully the data from our feasibility experiment will allow our group or another laboratory to be able to obtain support for a study of that type.

We respect the importance of the question about how long functional activity and immune protection would be sustained and have thus modified the ending of the Conclusion paragraph.  The new test  on Lines 355-360 is excerpted below:

Further, having demonstrated the value of nonhuman primates for this type of investigation, it would be important to also examine the influence of the neonatal Fc receptor for IgG (FcRn) in infant monkeys, because it can affect how long the maternal antibody remains in circulation [59]. In humans, the half-life of maternal IgG is approximately 3 weeks, which accounts for the decline in maternal antibody and immune protection by 6-9 months postpartum [60].

  1. The authors should determine the subclass of antigen-specific IgG antibodies in mothers and infants.

Unfortunately, we cannot conduct an analysis of the IgG subclasses in the samples collected from this study because the aliquots were depleted during the prior testing.  Based on prevailing animal welfare regulations, the blood volume collected from the neonates was limited (< 2 mL) for humane reasons.  That volume of blood generates only 1 mL of serum after centrifugation. Testing the specimens in multiple dilutions and in duplicate determinations used up the available sera. In addition, we prefer to use only previously unthawed aliquots in the assays. Testing the levels and biological activity of each IgG subclass specific to SARS-CoV-2 will also require the refinement of new assays. The CSA kit used for quantifying IgG levels was developed to enable cross-species testing of human and nonhuman primate sera from many species (chimpanzee, baboons, rhesus and cynomolgus macaques, and vervet monkeys) with the same assay platform.  However, it quantifies only the IgG isotype. 

We do agree with Reviewer 3 that while it is known that IgG1 is the primary subclass of maternal antibody transferred across the placenta, there could be an important reason to assess other IgG subclasses following immunizations of pregnant females, especially if the vaccines were administered in late gestation.

To address this concern, we have added the following text to the Concluding paragraph of the Discussion. In addition, we added a citation to a review that provides a comprehensive analysis of this topic.  Lines 407-416

Having verified that the rhesus monkey can provide an appropriate animal model for investigating the transfer of protective maternal antibody, future studies can address other critical issues, including vaccine safety [70], the optimal timing of maternal immunization for protecting preterm and term infants [71] and refinement of adjuvant formulations for use with recombinant protein vaccines. In addition, while studies have indicated that the placentally transferred antibody is predominantly IgG1, it will also be important to conduct a subclass analysis, especially after immunizations during pregnancy, because influenza vaccinations in late gestation were found to differentially affect the transfer rate of the IgG3 subclass to infants [72].  

Clements ,T.; Rice,T..F.; Vamvakas, G.; Barnett, S.; Barnes, M.; Donaldson, B.; Jones, C.E.; Kampmann, B.; Holder, B. Update on transplacental transfer of IgG subclasses: Impact of maternal and fetal factors. Front .Immunol. 2020, 11, 1920. doi: 10.3389/fimmu.2020.01920.

  1. For the neutralization assay, it is essential to have a positive control known to have significant neutralizing activity.

Thank you for bringing to our attention that we had not clearly conveyed that there were positive controls in the ACE-2 neutralization assay.  We have now emphasized that there were two Positive assay controls in the Methods section, as well as Negative assay controls. In addition, we compare the results for the experimental sera with the binding inhibition of the negative and positive controls in the figure legend for Figure 3.

Following the manufacturer’s instructions, the assay diluent was used as the primary Negative assay control, run in duplicate determinations, and its low binding inhibition used as the reference point when calculating the percent inhibition exhibited by the monkeys’ sera. BI (%) is calculated with the following formula: 1- (Experimental Sample/Assay Diluent) x 100. To provide additional Negative Controls, we included the pre-immunization baseline samples from the adult females, as well as sera from 6 additional seronegative adult monkeys who had not been immunized or exposed to SARS-CoV-2. To serve as Positive Controls, our assay protocol included anti-SARS-CoV-2 spike RBD neutralizing antibody (Human IgG1, ACRO Biosystems, SAD-S35, Lot # S35-211VF1-VT run 1:100 and 1:1000, 1.0 µg/mL, 0.1 mg/mL, respectively), and a pooled human SARS CoV-2 national IgG positive standard (Frederick National Laboratory Lot # COVID-NS0109, run at 1:10, 1:100 and 1:1000). The Positive Controls were included to validate the assay’s performance and not used to calculate the binding inhibition of the experimental samples.

We have also the following text to the figure legend for Figure 3, which provides a better context for the level of binding inhibition exhibited by the maternal and infant sera. Lines 287-293

As a Negative assay control condition, non-immunized sera from 12 seronegative female monkeys were also tested and elicited minimal binding inhibition (from 0 to <10%). Further, the sera from the immunized mothers and their infants elicited more binding inhibition than the two Positive assay controls at the dilutions tested. For monkeys immunized with RBD-Fc containing Wuhan protein sequences, the range of BI values was 35.8-99.3; for those immunized with RBD-Fc containing Gamma protein sequences, the range was 26.3-86.8. A graph showing the ACE2 neutralization results for 6 viral variants, including Delta and Omicron, is provided in Figure S1.

Minor point:

Although the authors have already shown their work, it would be nice to have illustrations of the vaccine antigen construct and the immunization and sampling schedule. 

Thank you for this suggestion.  We have created a new graphic that provides this information.  It is Figure S1 in Supplementary Materials.  The reader is referred to this new illustration in the Methods section to see a visual representation of the experimental protocol, sample collection schedule, and recombinant RBD-fc fusion protein. A copy of this new illustration is also included in the longer response to all 3 reviewers.

Round 2

Reviewer 1 Report

Comments and Suggestions for Authors

The revisions provided by the authors are acceptable and improved the manuscript.

Reviewer 2 Report

Comments and Suggestions for Authors

The authors have appropriately addressed all my concerns. The manuscript now looks good for publication. 

Reviewer 3 Report

Comments and Suggestions for Authors

The authors appropriately responded to the reviewer's comments.